# OpenReview forum: "Jailbreaking LLMs' Safeguard with Universal Magic Words for Text Embedding Models"
_ICLR.cc/2026/Conference — Submitted to ICLR 2026_

### Official Review · Reviewer_rKMg · 2025-10-31

**Soundness:** 3
**Presentation:** 3
**Contribution:** 3
**Rating:** 6
**Confidence:** 3

**Summary:**

This paper investigates a new vulnerability in LLM that rely on text embedding models. The authors identify a strong bias in the embedding distribution and leverage it to design efficient algorithms for finding universal magic words, which are an adversarial suffixes that can manipulate embedding similarity. In this way, these magic words can bypass text-embedding-based safeguards by altering perceived similarity between harmful and benign text pairs. The paper proposes three methods (brute-force, context-free, and gradient-based), evaluates their efficiency and transferability across models and languages. Moreover, the authors suggest renormalization-based defenses that improve robustness without retraining.

**Strengths:**

1. The paper studies vulnerability from an interesting angle by finding the universal magic words.
2. The paper shows strong attack and defense results for both the attack and the defense strategy.
3. The paper is well motivated and well written.

**Weaknesses:**

1. There lack of analysis on the possible number of magic words existing in a model.
2. The influence of repetition count, token length, or embedding normalization choices is not systematically analyzed.
3. There lack of analysis on the randomness in learning magic words across different random seeds, etc.
4. There lack of discussion on the origin/root/insights of the identified magic words.

**Questions:**

See weakness.

---

> ### Author Response · Authors · 2025-12-01
>
> Thank you for your helpful and positive comments. We address the raised points below.
>
> 1. Number of magic words
>
> Table 5 already gives an upper bound. The best magic words are usually found within the first two hundred or three hundred candidates, so the number of effective magic words is limited.
>
> 2. Repetition count and token length
>
> When we repeat a magic word or extend it with extra tokens, we observe a small initial gain but fast saturation. We will add this empirical observation.
>
> 3. Normalization choices
>
> In our later work, we tested several normalization settings. Removing the projection on the mean direction and then normalizing gives the best performance. A theoretical explanation is given there based on noise suppression. We will include a short note in the revision.
>
> 4. Randomness and random seeds
>
> Random seeds only change the order in which candidates appear in Table 5. They do not change which magic words are finally identified as the strongest. We will clarify this.
>
> 5. Origin of magic words
>
> Appendix F discusses this point. Magic words are extremely rare tokens. Rare tokens receive few updates during training, so some keep initial values that lie near the vulnerable direction. We will make this explanation more visible.
>
> Thank you again for the constructive feedback.

---

### Official Review · Reviewer_vDP5 · 2025-10-31

**Soundness:** 3
**Presentation:** 3
**Contribution:** 2
**Rating:** 4
**Confidence:** 3

**Summary:**

The paper shows that many text-embedding models have a strong mean-bias in their vector space and that short, universal “magic-word” suffixes can push any input toward this bias direction, manipulating cosine similarities that underpin embedding-based safeguards. Building on this observation, the authors present efficient search procedures—including a context-free black-box method aligned to the bias and a one-step white-box gradient approach—to find transferable suffixes. They demonstrate end-to-end jailbreaks by appending these words to user prompts and by requiring the model to end its responses with the same words, thereby bypassing both input and output guards. Experiments across multiple embedding backends and safety detectors report large drops in detection performance and cross-model transfer. To mitigate, the paper proposes a simple, train-free fix—mean-centering plus renormalization of embeddings—which substantially restores guard performance.

**Strengths:**

* this paper proposes a bias-direction analysis for text-embedding models, which is new
* this paper offers a simple, train-free mitigation for the proposed attack

**Weaknesses:**

* the usage of "magic" suffix has been proposed in other works (like GCG), making the contribution of this paper a bit incremental
* using renormalization for defense is promising but its impact on diverse downstream retrieval/semantic tasks (beyond the reported classifiers) remains underexplored
* lacks head-to-head experimental comparison with other whitebox attacks
* some inherited limitations of whitebox attacks

**Questions:**

* Could you please explain the novelty of the propose methodology comparing to other similar attacks? e.g.: https://arxiv.org/abs/2307.15043, https://people.eecs.berkeley.edu/~daw/papers/iris-naacl25.pdf
* Some papers question the transferability/generalizability of these universal adversarial triggers (e.g.: https://arxiv.org/abs/2404.16020v1). Under what conditions do your universal suffixes fail, and how does that compare to the existing analyses?

---

> ### Author Response · Authors · 2025-12-03
>
> Thank you for your detailed and constructive comments. We address the raised points below.
>
> **1. What's different from existing universal adversarial suffixes? Lacks comparison with other whitebox attacks.**
>
> The key novelty of our work is that we first time identify the distributional bias in text embeddings and leverage this phenomenon to find universal adversarial suffixes for LLMs' **safeguards** based on text-embedding. This focus is fundamentally different from prior white-box attacks and existing universal adversarial suffixes, such as [1,2], which attack **LLMs** rather than safeguards. Because our method attack entirely **different targets** (embedding-based safegurads instead of LLMs), head-to-head quantitative comparison with those existing attack methods would not be meaningful.
>
> We have cited those prior works (e.g., [1]) in Sec. 2.2 (lines 122–126). We appreciate your pointer to [2]. We will include it in the revision and add a clearer explanation of how our target, methods and contributions differ from those works.
>
> [1] https://arxiv.org/abs/2307.15043
>
> [2] https://people.eecs.berkeley.edu/~daw/papers/iris-naacl25.pdf
>
> **2. Inherits limitations of whitebox attacks. Generalizability of suffixes may sometimes fail.**
>
> We agree that existing studies have shown universal adversarial suffixes may not transfer across all LLMs, particularly those trained with strong preference-optimization techniques such as RLHF or DPO. However, our study targets embedding models rather than generative LLMs. To the best of our knowledge, no prior work has reported a similar lack of universality for attacks against embedding models. Understanding under which conditions such suffixes may fail in the embedding setting is an important open question, and we plan to explore it in future work.
>
> **3. Impact on downstream tasks remains underexplored.**
>
> Thank you for pointing out this important question. In our later work, we evaluated 38 text embedding models on a wide range of retrieval and semantic tasks on MTEB. The performance does not drop and often improves. This observation is also consistent with prior works such as arxiv 2103.15316. These results suggest that the proposed renormalization is safe for diverse downstream tasks and can even strengthen the representations. We will add a short clarification to the revision.

---

### Official Review · Reviewer_LP3e · 2025-11-01

**Soundness:** 3
**Presentation:** 2
**Contribution:** 2
**Rating:** 4
**Confidence:** 2

**Summary:**

This paper introduces a novel attack vector against LLM safeguards that are based on text embedding models. Appending magical words can fool the safeguard, i.e., classifiers trained on text embedding to distinguish harmful and harmless prompts. The core of the work is a key observation: the output distribution of several popular text embedding models is highly anisotropic, concentrating in a specific "band" on the unit hypersphere. The authors formalize this bias by identifying a "bias direction" (e*), which is the normalized mean of a large corpus of text embeddings. The paper then proposes defense methods against such attacks by fixing the defect of uneven embedding distribution.

**Strengths:**

The discovery and empirical validation of the non-uniform, biased distribution of text embeddings (Fig. 1) is a significant and insightful contribution. It provides a principled and elegant explanation for the existence of universal adversarial attacks, moving beyond simple heuristics. This observation itself is of high value to the representation learning. The paper is well-written and the finding on embedding is interesting. The authors also propose different methods driven by their finding.

**Weaknesses:**

* Does correcting the bias harm the embedding model's performance on its primary tasks (e.g., semantic search, classification)? An empirical evaluation is necessary. The setting on bypassing safeguard may also not be so useful for real-world applicability.
* The final step of Alg. 3 involves a Cartesian product of candidate tokens, which can lead to a combinatorial explosion. The practical limits on the magic word length and candidate size should be discussed.
* The defense method has not been tested for adaptive attacks, such as whether the method can defend against changes in jailbreakers.
* In Alg. 3, the comment "empirically better than zeros(h,m)" for random initialization is an interesting detail. A brief sentence of intuition would be helpful for the reader.

**Questions:**

N/A

---

> ### Author Response · Authors · 2025-12-01
>
> Thank you for your detailed and constructive comments. We address the raised points below.
>
> **1. Performance of primary tasks after correcting the bias**
>
> In our later work, we evaluated 38 embedding models after removing the mean projection. Their performance on semantic tasks does not drop and often improves. Prior work (https://arxiv.org/abs/2103.15316) also suggests this effect. We will add this clarification.
>
> **2. Practical limits and combinatorial explosion in Algorithm 3**
>
> Algorithm 3, the gradient-based method, is designed precisely to avoid the combinatorial explosion in the Cartesian product step. In the white-box setting, it optimizes multiple tokens simultaneously and does not need to enumerate combinations. We will state this more clearly.
>
> **3. Adaptive attacks**
>
> Since we propose a fix and hope practitioners adopt it, we do not encourage further development of adaptive attacks. We will add a short explanation of this concept.
>
> **4. Intuition for random initialization**
>
> Random initialization allows the method to probe the model with many small perturbations. After several steps of optimization, the noise averages out and the useful signal remains. Zero initialization is far from the token embedding distribution and does not provide this averaging effect.
>
> **5. Real-world applicability**
>
> Our main goal is to reveal an interesting property of current text embedding models and to encourage developers to close this security gap. As a byproduct, the same fix also improves embedding quality. We will make this motivation more explicit.
>
> Thank you again for the valuable feedback.

---

### Meta-Review · Area_Chair_rm7c · 2026-01-09

**Summary:**

Reviewers agreed the paper contains an interesting and potentially valuable observation about anisotropy / mean-bias in text embedding spaces, and demonstrated that short universal “magic-word” suffixes can bypass embedding-based safeguards. However, the main concerns were that the overall contribution felt incremental relative to prior universal suffix / trigger work, with limited head-to-head positioning against existing attack baselines, and that several key claims lacked supporting analysis—especially the impact of the proposed mean-centering / renormalization defense on core embedding tasks beyond the reported classifier-based safeguards. In addition, reviewers raised questions about practicality and completeness: e.g., limits of the proposed search procedure (potential combinatorial costs), whether the defense holds under adaptive attackers, and missing analysis on the diversity/quantity/origin/stability of discovered “magic words.” Taken together, these issues limited confidence that the work, as currently substantiated, meets the bar for acceptance despite the interesting core finding, motivating the final reject decision.

**Reviewer Concerns:**

The rebuttal partially addressed several important points: the authors clarified that subsequent experiments on many embedding models (including MTEB-style tasks) suggest mean-projection removal does not degrade and may improve semantic performance; they also clarified that the gradient-based method is intended to avoid combinatorial explosion, and provided intuition for random initialization. They additionally offered some discussion of repetition/length saturation and seed robustness. However, several concerns remain outstanding for this submission: the downstream-task evaluation is referenced as “later work” rather than being incorporated as concrete evidence in the paper; comparisons and positioning against related universal trigger/jailbreak methods remain largely qualitative (and the novelty still reads as incremental to some reviewers); and the paper still lacks deeper analysis of failure cases, adaptive attackers, and systematic characterization of magic-word frequency/structure beyond partial appendix discussion. Overall, while the rebuttal improves clarity, it does not fully close the gaps that drove the committee’s decision.

**Reviewer Scores:**

Reviewer LP3e (4→4): Likely to keep the score unchanged; they may be somewhat reassured by the claim that mean-centering does not harm semantic tasks and by clarifications around Alg. 3, but the lack of in-paper empirical validation for downstream performance and adaptive robustness would still keep them marginally below threshold.
Reviewer vDP5 (4→4 or 5): Might increase slightly (to a weak accept) due to clearer novelty framing (targeting embedding-based safeguards rather than LLM generation) and the added justification that defense is safe for retrieval tasks, but lingering concerns about incremental contribution and missing baseline comparisons would likely prevent a strong upgrade.
Reviewer rKMg (6→5 or 6): Could remain positive but might soften slightly in discussion given that several requested analyses (systematic characterization, stability/failure modes, deeper insight into magic words) are only partially integrated, and the rebuttal relies on follow-up work rather than fully demonstrated results in the submission.

---

### Decision · Program_Chairs · 2026-01-26

Reject